# Mechanical Recycling and Its Effects on the Physical and Mechanical Properties of Polyamides

**DOI:** 10.3390/polym15234561

**Published:** 2023-11-28

**Authors:** Ichrak Ben Amor, Olga Klinkova, Mouna Baklouti, Riadh Elleuch, Imad Tawfiq

**Affiliations:** 1Laboratoire QUARTZ EA7393, ISAE-Supméca Institut Supérieur de Mécanique de Paris, 93400 Saint-Ouen, France; ichrak.ben-amor@edu.supmeca.fr (I.B.A.); imad.tawfiq@isae-supmeca.fr (I.T.); 2Laboratoire des Systèmes Electromécaniques (LASEM), Ecole Nationale d’ingénieurs de Sfax, Sfax 3038, Tunisia; mouna.bakloutikharrat@gmail.com (M.B.); riadh.elleuch@gnet.tn (R.E.); 3Faculté des Sciences de Gafsa, Université de Gafsa, Gafsa 2112, Tunisia

**Keywords:** mechanical recycling, polyamide 6, polyamide 66, mechanical characterization, microscopic analysis

## Abstract

The aim of this study is to investigate the impact of mechanical recycling on the physical and mechanical properties of recycled polyamide 6 (PA6) and polyamide 66 (PA66) in relation to their microstructures. Both PA6 and PA66 raw materials were reprocessed six times, and the changes in their properties were investigated as a function of recycling number. Until the sixth round of recycling, slight changes in the mechanical properties were detected, except for the percentage of elongation. For the physical properties, the change in both flexural strength and Young’s modulus followed a decreasing trend, while the trend in terms of elongation showed an increase. Microscopic analysis was performed on virgin and recycled specimens, showing that imperfections in the crystalline regions of polyamide 6 increased as the number of cycles increased.

## 1. Introduction

Polymer recycling is the process of recovering and reusing different types of plastics that would otherwise end up in landfills or in the oceans, polluting the environment. Polymer recycling involves the use of advanced technologies to turn plastic waste into new raw materials that can be used to manufacture new products [1,2]. Polymer recycling is essential for maintaining environmental balance, reducing waste and pollution, and protecting natural resources. It also helps reduce the carbon footprint and energy consumption associated with the production of new plastics [3].

Recycling processes involve collecting plastic waste from homes, offices, and industrial factories, sorting and cleaning it to remove contaminants, grinding it into small pellets, and then melting it down to make new products [4,5]. There are different types of polymers recycling processes, including chemical recycling, biodegradable polymer recycling, and mechanical recycling [5,6,7]. Chemical recycling uses techniques such as pyrolysis and gasification to break down plastic waste into its component parts and then convert them into new materials [8]. The recycling of biodegradable polymers uses biodegradable plastics made from renewable raw materials such as corn or potato starch, which, under certain conditions, can break down into organic substances, water, and carbon dioxide [6,7,8,9]. Mechanical recycling is the processing of plastic waste by crushing and grinding it into small granules that can then be melted down and formed into new products [10]. 

Polyamide, commonly known as nylon, is a synthetic polymer with a wide range of applications in different industries. It is widely used in several industrial applications due to its exceptional durability, strength, and compatibility with various manufacturing techniques, making it an essential material for many industrial processes. Some of the major industrial applications of polyamide include the automobile, textile, and packaging industries [11,12]. The automobile industry uses polyamide material extensively for making various parts such as engine covers, air intake manifolds, and door handles. This is because it has excellent heat and chemical resistance, can withstand high loads and forces, and is lightweight. Similarly, the textile industry commonly uses nylon for making sportswear, hosiery, and swimwear due to its moisture-wicking capacity, high tensile strength, and abrasion resistance [11,12,13,14].

Moreover, polyamide can be either mechanically recycled or chemically recycled through thermal depolymerization [15]. Mechanical recycling involves grinding down the used polyamide products and then melting them to form new products, while chemical recycling involves breaking down the used polyamide into its constituent molecules and then reassembling them into new products.

PA’s recyclability makes it a sustainable and environmentally friendly material by reducing the amount of waste in landfills, reducing the carbon footprint of manufacturing, and reducing the cost of production since reprocessing old materials is cheaper than producing new ones [1,2]. Despite these promising benefits, reprocessing polyamide can lead to a reduction in its molecular weight [16], which can affect its mechanical properties, and reduce its thermal stability, making it more susceptible to melting or deformation and to color alteration due to contamination, which can reduce the aesthetic aspect of the end product.

Polyamides find applications in diverse environmental conditions, experiencing fluctuations in temperature and humidity over time. Given its hygroscopic nature, PA6 has the capacity to absorb between 9.5 and 10% of its weight in moisture, and PA 66 is capable of absorbing up to an average of 8–10% of water upon saturation [17,18,19]. The presence of water into polyamide is influenced by factors like temperature, loading conditions, and time. The introduction of moisture can enhance the flexibility of the PA6 chains, leading to changes in its mechanical properties. One consequence of this moisture absorption is the observed reduction in the material’s effective stiffness, as evidenced by experimental findings [20,21]. Thirumalai et al. conducted research on the use of PA-6 as a polymeric matrix in fiber composite wind turbine blades [22]. They reported that PA-6 absorbed approximately 3 wt.% of water at 296 K and 50% relative humidity. The time required for PA-6 to reach moisture equilibrium was found to be contingent on the thickness of the specimen.

Gnanamoorthy et al. [23] conducted a study where they fabricated a melt extrusion nanocomposite comprising PA6 and organically modified hectorite. They observed that heightened ambient humidity and water uptake led to a reduction in both hardness and modulus, while simultaneously enhancing the flexural fatigue life. In a separate investigation by Krzyzak et al. [24], the influence of varying amounts of glass fiber in a PA6-based composite was examined. This study specifically delved into the moisture absorption from the atmosphere at 278 K and 70–80% relative humidity. The findings indicated that the inclusion of 20% and 30% of glass fiber resulted in approximately a two-fold and three-fold decrease in water absorption for the pellets immersed in water. A meticulous drying process was undertaken in accordance with the supplier’s specifications in this study to prevent excessive moisture in PA6 and PA66, especially during recycling.

The present paper focuses on assessing the impact of multiple cycles of recycling on the physical–mechanical properties and microstructure of PA6 and 66 when subjected to injection molding. The objective is to gain insights into the number of times PA6 and PA66 could undergo recycling without compromising their various properties, thereby producing a sustainable and useful material. The materials studied are mechanically recycled, which is one of the most common methods used to recycle PA, as it enables the material to be transformed into a range of products, such as automotive parts, electronic components, and household appliances [25]. However, it is crucial to ascertain the effect of multiple recycling cycles on the quality and durability of the material before widespread use. Therefore, this paper will delve into the experimental results obtained after subjecting polyamide 6 and polyamide 66 to six cycles of injection molding and provide insights on the effect of multiple recycling steps on the physical–mechanical properties and morphology of the materials dedicated to the manufacturing of assembly components in a finished product. Hence, their UV degradation has not been investigated.

## 2. Materials and Methods

### 2.1. Materials

The materials under investigation in this study consist of two types of polyamide granules: polyamide 6 (Teknor Apex, Rothenburg ob der Tauber, Germany, Chemlon MD82) and polyamide 66 (Technyl Solvay, Leuna, Germany, A205F BLACK 21N). The black color of both types of granules is achieved by incorporating 2% of dye pellets, also known as “color masterbatch”, into a batch of white-colored base material granules. This process enables the uniform dispersion of the dye throughout the granules, ensuring consistent color quality and reproducibility, resulting in higher sample quality and ensuring opacity is preserved throughout all the process steps. The use of masterbatch in the plastic industry is a well-established technique for producing high-quality granules with optimal physical and chemical properties for specific applications [26]. Table 1 presents the properties and essential attributes for the injection of the PA6 and PA66 materials used in this study.

### 2.2. Sample Preparation

The experimental procedure entailed six cycles of mechanical recycling. Each cycle within the series involved a sequence of grinding, followed by a drying process, and concluded with injection molding. At various intervals within each series, samples were extracted for subsequent analysis. The rectangular specimens used in this study were produced using the Kraussmafei 50-ton injection molding machine. The injection conditions provided by the technical documents of the producers are summarized in Table 2. The specifically designed mold was used to create the specimens illustrated in Figure 1a, with final dimensions of 125 × 13 × 3 mm^3^. Polyamides are known for their hygroscopic nature. This property can significantly impact the quality and properties of the final product, particularly in melting processes. Additionally, it synergistically interacts with thermomechanical degradation, further accentuating the loss of polyamides’ properties. Prior to injection, the PA6 material was carefully dried for 2 h at 80 °C and PA66 was dried for 4 h at the same temperature to remove any moisture that may have accumulated during storage or handling. This step is critical to ensure that the material properties are consistent and reproducible throughout the testing process. The dimensions of each specimen were precisely controlled. Some of the specimens of the first batch were then granulated in a grinding mill to obtain pellets of a uniform size (2.8 mm in mean diameter). These pellets were then carefully dried and re-injected under the same conditions as the virgin PA. The resulting specimens were tested and analyzed again. This recycling process, starting with the collection of sprues, milling, drying, injecting, and assessment, was repeated six times for both PA6 and PA66.

Once all the rectangular plates were molded (Figure 1b), the standard tensile samples (Figure 1c) were cut. The tooling was conducted using a CIF Techno-drill 3 XL milling machine with a 2.5 mm diameter bur, an advancing speed of 5 mm/s, a rotation frequency of 22,000 rpm, and a cutting depth of 1 mm.

### 2.3. Materials Characterization

#### 2.3.1. Material Characterization

Quasi-static tensile and 3-point bending tests were performed on the studied materials using the Zwick/Roell Z100 universal machine. The tensile tests were conducted at room temperature using a 100 kN force cell and Clip-On Extensometer strain gage following the ISO 527-4 standard [28] (Figure 2a), with a test rate of 2 mm/min. 

Flexural tests (Figure 2b) were conducted on the samples in accordance with the ASTM D-790 standard [29], with dimensions of 125 × 13 × 3 mm^3^ on a Zwick/Roell Z100 tensile machine with a 100 kN force cell. The tests were run at room temperature with a test rate of 1 mm/min. Each measurement was repeated on 5 samples.

#### 2.3.2. Microscopic Analysis

Scanning Electron Microscopic (SEM) observations were conducted using the JEOL JSM6010 PLUS/LA microscope. Transversally cut surfaces of cracked specimens resulting from tensile tests were not metallized and analyzed under low pressure to reduce the interference caused by the atmosphere (Figure 3). In order to obtain a comprehensive understanding of the damage mechanics, the SEM was performed on virgin and polyamides recycled six times to accurately compare the degradation in material quality at different levels of use.

#### 2.3.3. Fluidity Test

The comprehensive fluidity test was carried out in accordance with the ISO 1133-1 standard [30]. In this test, PA6 and PA66 melt flow rate measurements were performed, namely the material flow index measured by mass (MFR) and the material flow index measured by volume (MVR).

For MFR, the mass of material extruded in a given time is measured (expressed in g/10 min). For MVR, the volume of material extruded in a given time is measured (expressed in cm^3^/10 min). MVR is perhaps the preferred measurement of flow behavior due to its independence of density, especially for plastics with rheological properties that are sensitive to their moisture content, such as polyamide [16]. A standard die was employed, with a nominal length of 8 mm and a nominal bore diameter of 2 mm. The specific conditions used for each material are summarized in Table 3.

## 3. Results

### 3.1. Tensile Test

Moisture plays a crucial role in influencing the mechanical properties of polyamides. To mitigate this effect, the materials were carefully maintained at a moisture content under 0.2%. The accurate measurement of moisture levels was achieved using the MS-70 Moisture Analyzer from A&D. In this study, PA6 underwent six reprocessing cycles. Interestingly, no significant color change was observed in either PA6 or 66 between the virgin material and the material subjected to six cycles. Figure 4 illustrates the correlation between the number of processes and the mechanical characteristics of both PA6 and PA66.

In our investigation, we conducted tests on virgin PA6 and PA66 to assess its tensile modulus and tensile strength. The results were then compared with the data provided by the supplier in Table 1 of the technical datasheet. For virgin PA 66, our test results closely align with the values specified by the supplier. The measured parameters exhibit a high level of similarity, indicating a strong consistency between our experimental findings and the manufacturer’s provided data. However, a notable difference was observed for PA6, particularly in the tensile strength parameter, where our measured value was 51 MPa, while the supplier’s datasheet indicated a value of 70 MPa. Several factors could contribute to this observed difference. Firstly, the raw material used to manufacture polyamide 6 may exhibit variations between different batches. These variations can arise due to differences in the properties of the raw materials, even if they comply with the supplier’s specifications. Secondly, the manufacturing process, including factors such as temperature, pressure, and cooling times, can influence the mechanical properties of the material. Deviations from the specified conditions during the production process may lead to variations in the material’s performance. Eventually, the effects of storage conditions on the material cannot be discounted. Changes in environmental conditions, such as humidity or temperature during storage, may impact the material’s properties over time.

Young’s modulus decreased by up to 20% from the virgin state to the first cycle for PA6, from 2490.16 to 1996.16 MPa. Then, it reduced slowly by about 5% as the number of processes augmented. At the sixth cycle, it decreased to 1595.76 MPa (Figure 4a). As illustrated by Young’s modulus, a decreasing pattern was also observed in tensile strength. Figure 4b shows that it was reduced by 14% after the sixth reprocessing cycle. Moreover, alterations in elongation (%) were noticed with each process. A considerable increase in elongation (%) of about 21% occurred between the virgin PA6 and its first recycling cycle, followed by successive increases, whose intensity depended on the number of recycling cycles. Cycles 2 and 3 exhibited similar elongations (%). After the third cycle, the improvement was more pronounced, and reached 22% at the sixth cycle. In brief, a total increase of 49% in elongation (%) was observed (Figure 4c).

The variation in the mechanical properties of polyamide 6 has already been investigated by Maspoch et al. [31] and Crespo et al. [32] using PA6 scrap obtained from injection molding. They demonstrated that with an increase in the number of processing operations, Young’s modulus and the maximum tensile stress decreased for PA6. Additionally, for more than three processing cycles, a reduction of 28% in tensile stress and 11% in Young’s modulus was observed.

Meanwhile, in the case of PA66, there was an 18% decrease in Young’s modulus from the virgin state to the first cycle, followed by a slow and continued reduction up to the sixth cycle. In total, 34% of the reduction was unregistered compared to the virgin material in the sixth cycle (Figure 4a). As demonstrated by the tensile modulus, tensile strength also showed a decreasing trend. It was reduced by 64% after the sixth reprocessing cycle. Additionally, changes in elongation (%) were observed throughout each process. Similar to PA6, between virgin PA66 and the first recycling cycle, there was an improvement in elongation (%) of 16%, while a subsequent increase of about 20% was identified between the virgin and the second recycling cycle. To conclude, an overall increase of 52% in elongation (%) was noted. These results align with the findings of Djeddi and Mohellebi [33] who demonstrated that polyamide PA66 exhibits strong mechanical resistance characteristics with a low elongation at break. Recycling this material leads to a notable improvement in ductility, around 233%, accompanied by a reduction in maximum stress by 11%.

#### Microscopic Analysis

SEM images were utilized to examine the inclusion size alterations in both virgin and recycled PA6 and PA66. Using 500× magnification, microphotographs were captured to investigate the qualitative changes in inclusion size. Figure 5 shows the microstructure of the virgin PA6 at the cracked section in the situations of dried and non-dried material before injection. Three elements can be distinguished: cupules, inclusions, and the matrix. In the case of undried PA6, most cupules contain inclusions (Figure 5a). However, the difference for the dried PA6 is that some cupules become empty and the inclusions disappear, or their sizes are reduced [Figure 6a–g, Table 4], potentially due to the formation of non-melted granules arising from polymer degradation that contributes to the development of smaller crystals [34], leading to a higher crystallization temperature. Consequently, with a higher number of recycling cycles, inclusion sizes become smaller, displaying changes in the crystalline regions [35].

The investigation of the impact of recycling on microstructure is shown in Figure 6. Nevertheless, the distinction between differences seems insufficient based on simple observations. This is why we used the image analysis tool Image J v1.53k software from Wayne Rasband and the US National Institutes of Health [36]. To analyze its pixels, this program transforms particle fractions into a binary image. Particle area fractions were obtained by counting pixels [37]. The results are given in Table 4. It was found that for PA6, the mean size of inclusion diminished as the number of recycling increased. 

The total area of inclusions on the treated SEM images also decreased continuously from the virgin state to the fifth recycling cycle. Also, the percentage of porosity increased as the number of recycling cycles increased. This modification of the microstructure’s composition was linked to the decrease in viscosity and mechanical properties observed previously. The degeneration of inclusions can be explained by chain breakage or hydrolysis. Consequently, with a higher number of recycling cycles, the inclusion sizes became smaller and the porosity of the material became higher.

As shown in Figure 7, SEM analysis for polyamide 66 was carried out to determine the mode or cause of failure. The fracture morphology of polyamide 66 had a brittle fracture morphology since the first cycle. This was due to contamination particles that produce these defective sites.

### 3.2. Flexural Test

The flexural properties are summarized in Figure 8. Regarding virgin PA6 and PA66, a slight difference has been observed in the results for both the flexural modulus and flexural strength presented in Table 1. This variance is attributed to several factors mentioned previously and potential effects of injection molding, where the direction of the material injection is perpendicular to the direction of the applied flexural force. This perpendicularity creates a unique stress distribution within the material, impacting its response to bending. Consequently, the mechanical properties measured in flexural testing may exhibit variations compared to the specifications provided by the supplier.

The flexural strengths of virgin polyamide 6 and polyamide 66 were 156.06 MPa and 192.80 MPa, respectively, which were reduced to 99.72 MPa and 113.80 MPa after the sixth reprocessing cycle. The changes in flexural strength and flexural modulus both showed a decreasing trend, while the change in elongation showed an increasing trend. The trends of the results are similar to those of the tensile strength properties discussed above. This conclusion aligns with the research conducted by Mospoch et al. [31].

### 3.3. Fluidity Test

Figure 9 and Figure 10 illustrate the MFR and MVR of recycled PA6 and PA66. For PA6, they are linked to the increase in processing cycles until the fifth recycling cycle, signifying a decrease in viscosity due to chain breakage or hydrolysis. Starting from the sixth processing cycle, a notable reduction in MFR and MVR values is observed, which could be attributed to degradation caused by an increase in molecular weight due to crosslinking or condensation processes [16].

For PA66 and during the first recycling cycle, the variation in the melt index reveals fascinating dynamics. At this stage, the extrusion volume per unit of time increased while the mass decreased, implying a potential interaction between viscosity and porosity effects. It is noteworthy that the material exhibited increased porosity, demonstrated by a reduction in the melt flow rate (MFR), along with a decrease in viscosity, resulting in improved fluidity, as evidenced by a higher melt volume rate (MVR). As a result, the degradation caused by chain breakage became apparent in the first recycling cycle, affecting both MVR and viscosity.

In the following recycling cycles (cycles 2 through 5), MVR decreased again, falling below the original levels of the material. This caused an increase in viscosity due to degradation due to increased molecular weight and crosslinking. An intriguing change took place in the sixth recycling cycle, when the MVR trend reversed and displayed an average value higher than that of the fifth cycle. This reversal suggests that degradation through chain breakage reemerged as a contributing factor.

The degradation of the material, characterized by breaks in polymer chains leading to shorter chains, results in a decrease in viscosity. This phenomenon occurs when the material undergoes repeated processing, such as injection, extrusion, and palletization. This trend is consistent with findings reported by other researchers [31,32,38] who studied the reprocessing of PA6 and PA66. The viscosity of the materials diminishes with repeated cycles. However, comparing the final data is challenging due to various factors, including differences in the materials used, variations in process parameters, and potential contamination during the recycling process. These variables could influence the final results, making an exact comparison challenging.

## 4. Conclusions

Mechanical recycling significantly influenced the mechanical and physical properties of both PA6 and PA66, attributed to alterations in their molecular structure during the recycling process. Both polyamide 6 and polyamide 66 underwent multiple recycling cycles via injection molding, totaling up to six cycles. The main outcomes of the study concern the impact of recycling on mechanical characteristics and melt viscosity, evaluated through tensile, flexural, and fluidity tests. The summarized results are as follows:There was a noticeable decrease in mechanical properties for both materials. Specifically, Young’s modulus reduced by 36% and 34% for PA6 and PA66, respectively, from the virgin state to the sixth cycle. The tensile strength showed reductions of 14% and 64% for PA6 and PA66, respectively.There was a notable average increase of 50% in elongation (%) for both materials. In the flexural test, similar trends were observed as in the tensile strength properties. These variations can be attributed to factors like chain scission, molecular weight degradation, and the accumulation of impurities during the recycling process.Changes in the physical properties were evident in both materials, as demonstrated by the MFR and MVR measurements. These changes were linked to the increase in processing cycles, with MFR reaching 17.79 g/10 min and 5.51 g/10 min for PA6 and PA66, respectively, by the fifth recycling cycle. This indicates a reduction in viscosity likely due to chain breakage or hydrolysis, resulting in improved processability.The recycling process had an impact on the increase in the total area of porosity, reaching 33.1% for PA6. These alterations can have adverse effects on the material’s mechanical and other physical properties, leading to a decrease in performance over time.

## Figures and Tables

**Figure 1 polymers-15-04561-f001:**
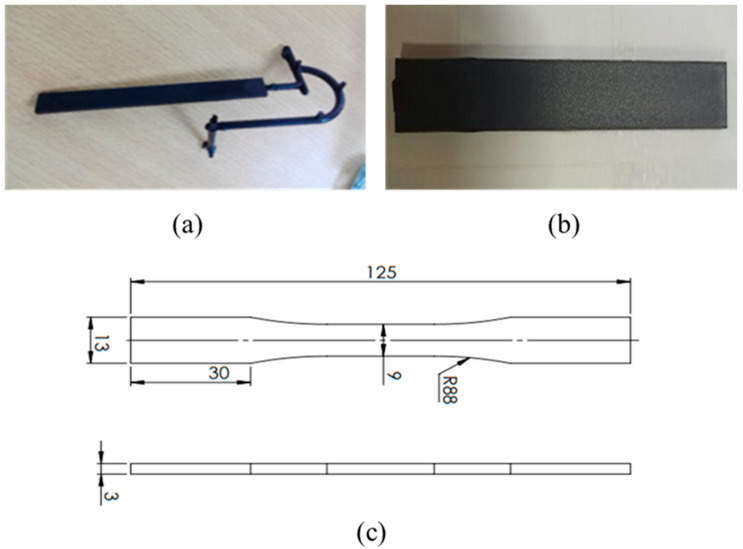
(**a**) Injected specimens; (**b**) rectangular specimen for cutting; (**c**) geometry of tensile test sample in mm according to ASTM D638 [27].

**Figure 2 polymers-15-04561-f002:**
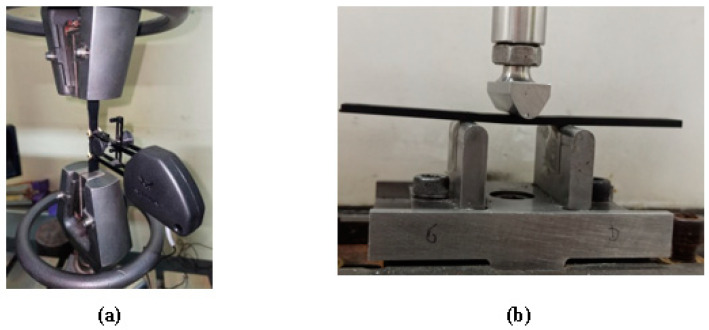
(**a**) Tensile test set-up; (**b**) 3-point bending test set-up.

**Figure 3 polymers-15-04561-f003:**
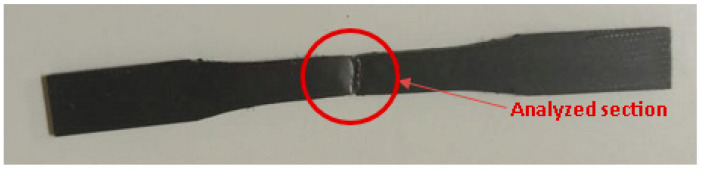
Cracked specimen.

**Figure 4 polymers-15-04561-f004:**
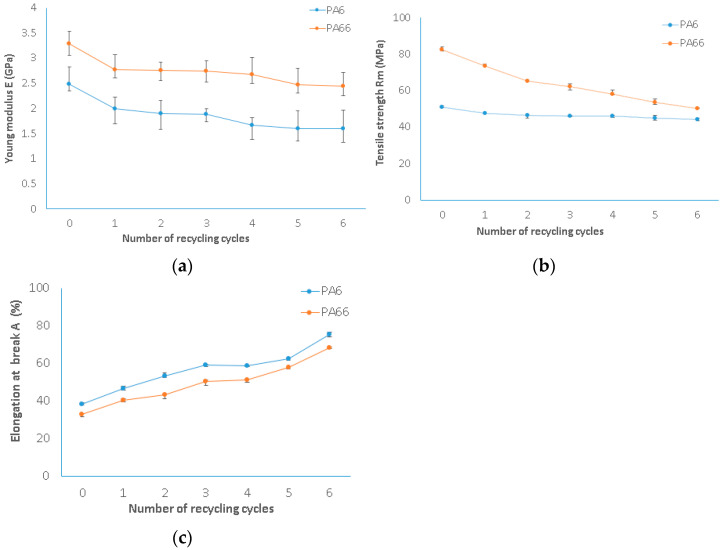
Mechanical characteristics of virgin and recycled PA6 (blue line) and PA66 (orange line): (**a**) Young’s modulus; (**b**) tensile strength; (**c**) elongation at break.

**Figure 5 polymers-15-04561-f005:**
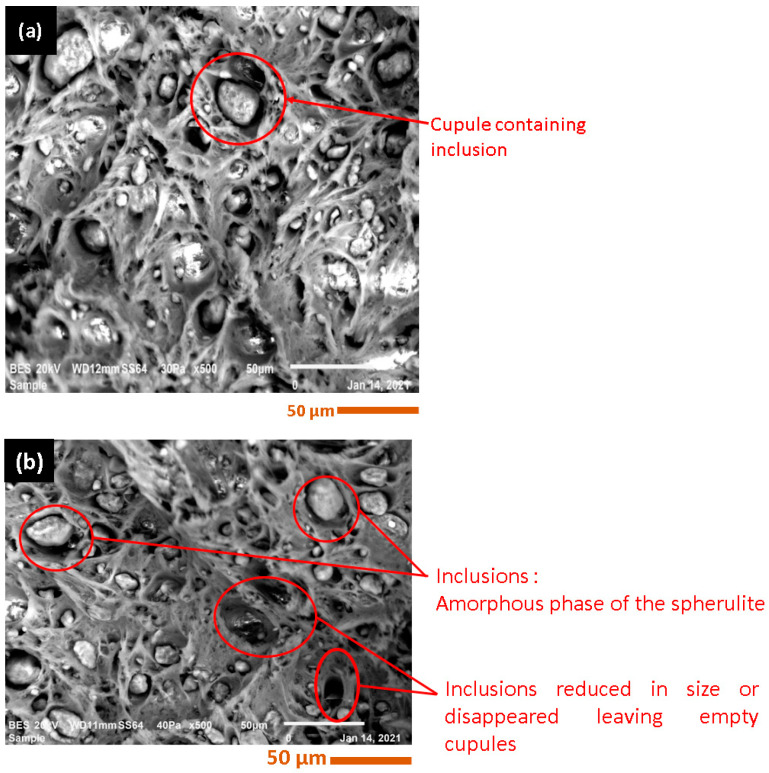
Rupture surface morphology of (**a**) virgin dried PA6; (**b**) virgin undried PA6.

**Figure 6 polymers-15-04561-f006:**
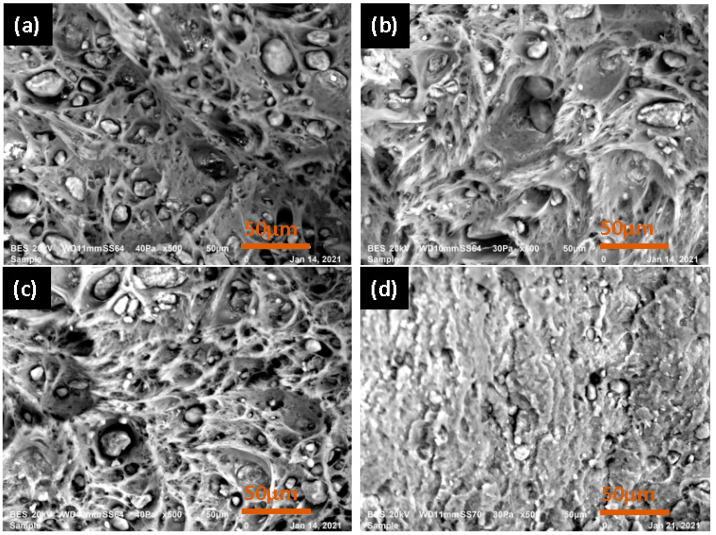
SEM images of: (**a**) virgin PA6; (**b**) first recycling; (**c**) second recycling; (**d**) third recycling; (**e**) fourth recycling; (**f**) fifth recycling; and (**g**) sixth recycling.

**Figure 7 polymers-15-04561-f007:**
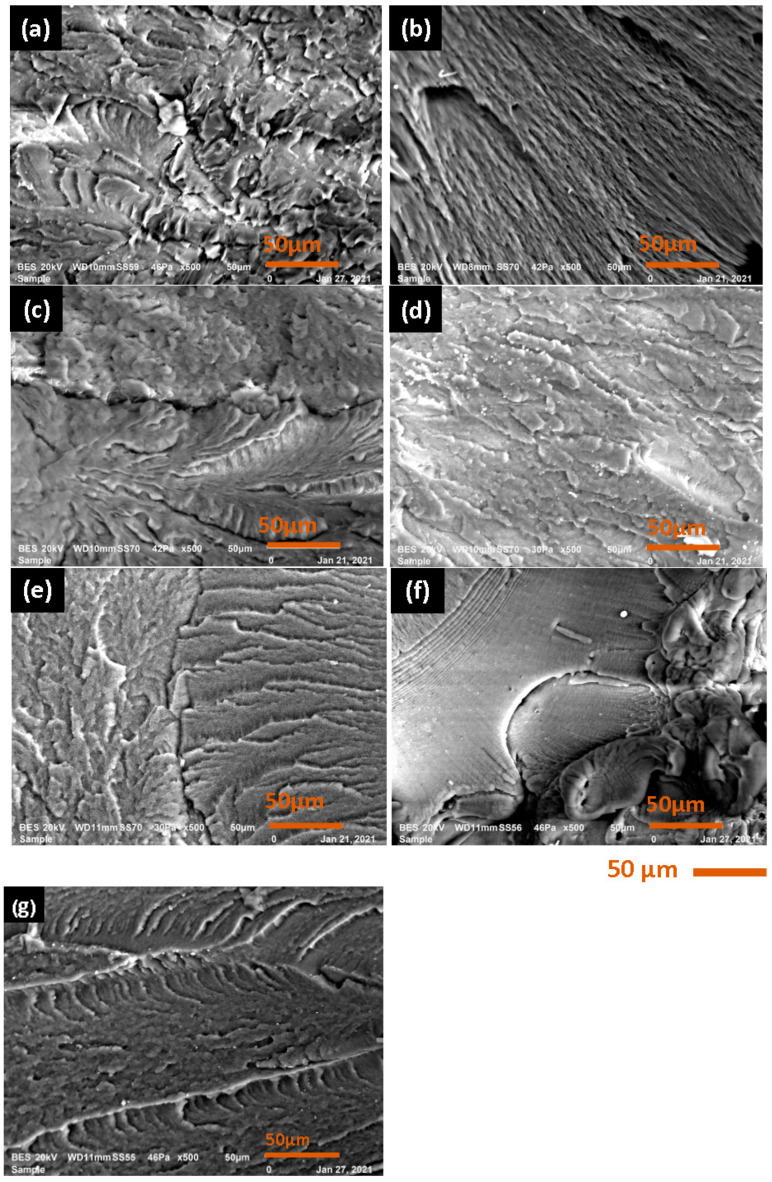
Surface morphology SEM images: (**a**) virgin PA66; (**b**) first recycling; (**c**) second recycling; (**d**) third recycling; (**e**) fourth recycling; (**f**) fifth recycling; and (**g**) sixth recycling.

**Figure 8 polymers-15-04561-f008:**
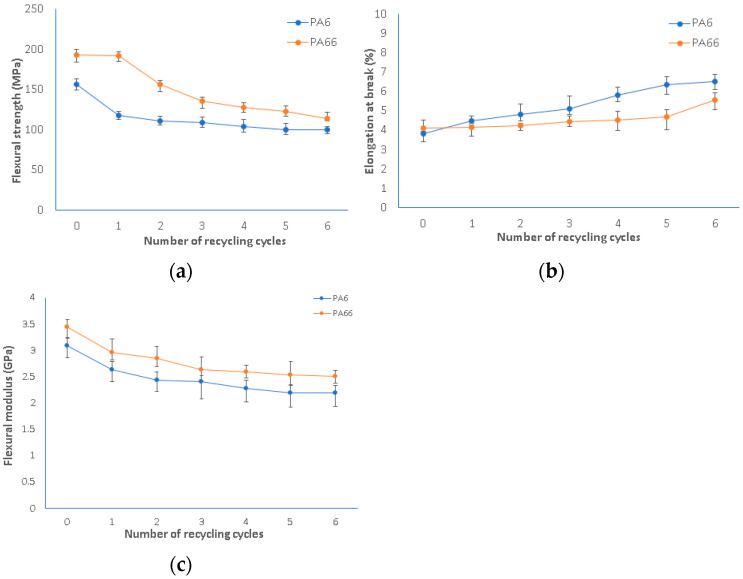
Flexural properties of virgin and recycled PA6 and PA66: (**a**) flexural strength; (**b**) elongation at break; (**c**) flexural modulus.

**Figure 9 polymers-15-04561-f009:**
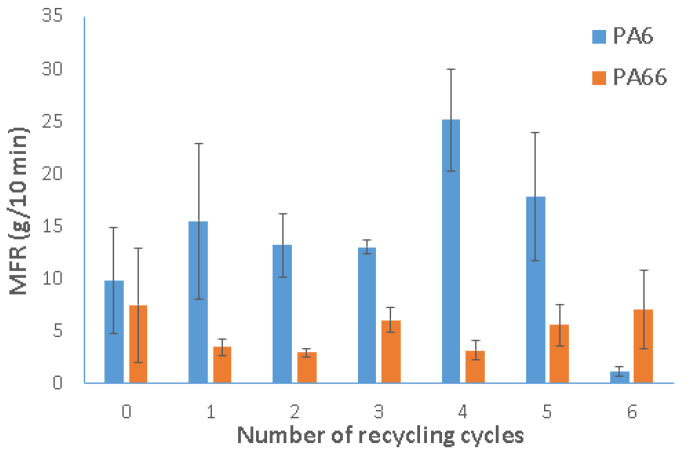
Hot melt indices of recycled polyamide 6 and polyamide 66, recycled from 1 to 6 times, by mass (MFR).

**Figure 10 polymers-15-04561-f010:**
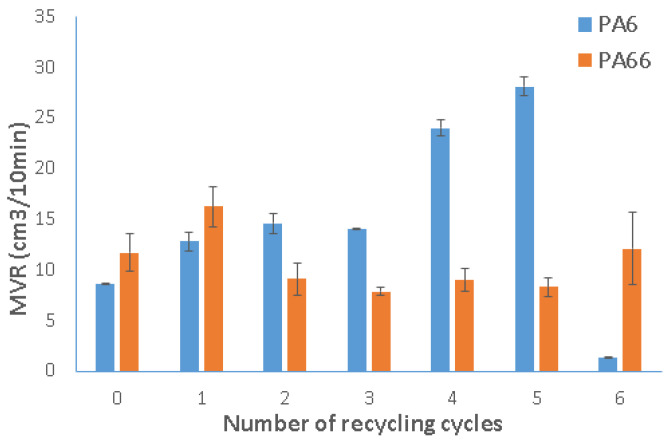
Hot melt indices of recycled polyamide 6 and polyamide 66, recycled from 1 to 6 times, by volume (MVR).

**Table 1 polymers-15-04561-t001:** Properties of PA6 and PA66.

Property	PA6	PA66
Density, g/cm^3^	1.18	1.14
Molding shrinkage, %	1.1–1.5	
Water absorption (Equilibrium, 23 °C, 50% RH), %	2.5	2.6
Tensile modulus, MPa	2900	3100
Tensile stress, MPa	70	82
Tensile strain, %	4.0	4.5
Flexural modulus, MPa	3100	2800
Flexural stress (strain de 3.5%), MPa	85	115
Heat deflection temperature 1.8 MPa, unannealed, °C	80	70
Coefficient of thermal expansion, cm/cm/°C	6.0 × 10^−5^	1.0 × 10^−4^
Thermal conductivity, W/m·K	0.34	0.16

**Table 2 polymers-15-04561-t002:** Injection molding parameters.

Injection Parameters, Units	PA6	PA66
Processing melt temperature, °C	240–260	280–300
Mold temperature, °C	60–80	50–90
Holding pressure, MPa	140	100
Back pressure, MPa	15	-
Screw Speed, m/min	30	24

**Table 3 polymers-15-04561-t003:** Fluidity test parameters.

Materials	Loads, kg	T, °C
PA6	1.2	235
PA66	0.325	275

**Table 4 polymers-15-04561-t004:** SEM images analysis vs. of number of recycling cycles of PA6.

Number of Recycling Cycles	Mean Area of Inclusion (μm^2^)	Total Area of Inclusion (µm^2^)	Area of Porosity (µm^2^)
0	68.3	5532.0 (10.2%)	6822.7 (12.6%)
1	52.3	3823.9 (7.1%)	7542.0 (14.0%)
2	42.8	3602.3 (6.6%)	9318.1 (17.3%)
3	37.2	2420.1 (4.5%)	10,374.0 (19.2%)
4	33.9	2376.8 (4.4%)	15,341.2 (28.5%)
5	27.3	2184.5 (4.0%)	17,719.2 (32.9%)
6	13.3	1122.8 (2%)	17,854.5 (33.1%)

## Data Availability

The data presented in this study are available on request from the corresponding author.

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
