# Peer review of "Mechanical Recycling and Its Effects on the Physical and Mechanical Properties of Polyamides"

_polymers, 2023, doi:10.3390/polym15234561_

Round 1

Reviewer 1 Report (Previous Reviewer 1)

Comments and Suggestions for Authors

The work has been well improved. It can be considered for publication.

Comments on the Quality of English Language

none

Author Response

Dear editor and dear reviewers,

Thank you very much for taking the time to review this manuscript. We deeply appreciate your insightful comments and inquiries. Please find the detailed responses below and the corresponding revisions/corrections highlighted in distinct colors to differentiate between the editor's and reviewers' input. Additionally, we have adjusted the title in accordance with the provided recommendations.

Best regards,

Olga KLINKOVA

Reviewer 2 Report (New Reviewer)

Comments and Suggestions for Authors

The abstract advances relevant results and makes some indications of the value of the research. Nonetheless, I would like to strengthen the novelty of the paper.

The introduction is well written discusses the main aspects and makes a sensible state of the art.

 In line 101, the authors can change from “several cycles of injection…” to the number of cycles they have made.

Table 1, the authors can evaluate changing from tensile modulus to Young’s modulus and from tensile and flexural stress to tensile and flexural strength.

Figure 1, cite the iso or ASTM standard for the specimens.

Do the authors use figure 1c specimens for the flexural test? Otherwise, define the flexural test specimens and the standard used.

State the number of specimens tested.

I suggest using GPa for the moduli.

Figure 4a and 8c in view of the standard deviations an ANOVA analysis can be done to stablish if the differences are statistically relevant.

My main complaint is that the paper presents the results but does not discuss such results. A more elaborate discussion of the results, comparing the behavior with other studies, and justifying the causes of the behavior of the tensile and flexural properties can increase the quality and interest of the paper.

Author Response

Dear editor and dear reviewers,

Thank you very much for taking the time to review this manuscript. We deeply appreciate your insightful comments and inquiries. Please find the detailed responses below and the corresponding revisions/corrections highlighted in distinct colors to differentiate between the editor's and reviewers' input. Additionally, we have adjusted the title in accordance with the provided recommendations.

Reviewer 2

  1. In line 101, the authors can change from “several cycles of injection…” to the number of cycles they have made.

Done (replaced by “six”).

  1. Table 1, the authors can evaluate changing from tensile modulus to Young’s modulus and from tensile and flexural stress to tensile and flexural strength

Done from line203 to line 220 and from line 211 to line 218.

  1. Figure 1, cite the iso or ASTM standard for the specimens.

Done.

  1. Do the authors use figure 1c specimens for the flexural test? Otherwise, define the flexural test specimens and the standard used.

It was noted in the text line 156, but we revised the sentence for improved it.

  1. State the number of specimens tested.

Done Line.

  1. I suggest using GPa for the moduli.

Done.

  1. Figure 4a and 8c in view of the standard deviations an ANOVA analysis can be done to stablish if the differences are statistically relevant.

We didn’t perform the ANOVA analysis because for all the tests, five repetitions for each test were done and presented, which is recommended by standards ISO 527 and ASTM 790.

  1. My main complaint is that the paper presents the results but does not discuss such results. A more elaborate discussion of the results, comparing the behavior with other studies, and justifying the causes of the behavior of the tensile and flexural properties can increase the quality and interest of the paper

Done:

  • line 232 to line 237
  • Line 247 to line 251
  • Line 324
  • Line 362 to line 370

Reviewer 3 Report (New Reviewer)

Comments and Suggestions for Authors

The authors have addressed the issue that has been of interest for number of years. Recycling of PA is an important topic in green chemistry. However, the following should be resolved:

1. Structural changes analyzed using FTIR should be included, to follow and connect to mechanical and morphological analyses.

Author Response

Dear editor and dear reviewers,

Thank you very much for taking the time to review this manuscript. We deeply appreciate your insightful comments and inquiries. Please find the detailed responses below and the corresponding revisions/corrections highlighted in distinct colors to differentiate between the editor's and reviewers' input. Additionally, we have adjusted the title in accordance with the provided recommendations.

Reviewer 3:

  1. Structural changes analyzed using FTIR should be included, to follow and connect to mechanical and morphological analyses.

Many researchers have highlighted the challenge of recycling polyamides and have utilized FTIR in their studies to analyze chemical structures. Despite their efforts, the chemical structure of polyamide remained unchanged in the FTIR spectra. As an example, in section 3.1 of this paper: https://doi.org/10.1016/j.jmatprotec.2007.04.056

Based on bibliography we choose not conduct FTIR analysis, as our research objective was centered around understanding how the recycling process influenced the mechanical and physical characteristics of the polyamide.

Reviewer 4 Report (New Reviewer)

Comments and Suggestions for Authors

Dear Authors,

Thanks for studying the mechanical properties of Polyamide based materials after several recycling processes. This may potentially help to improve them and, hopefully, increase the recycling culture. Manuscript is well described and deserves being published in Polymers' journal, after a careful consideration of the following comments:

- UV is known to affect polyamides' properties (for instance, https://doi.org/10.1007/s42452-020-03319-4). Were samples exposed to UV light between each cycle? If not, it is suggested to mention that samples were not exposed to UV light in the manuscript.

- Was hydrophobicity inspected between cycles? Perhaps inclusion and porosity modification on the surface alters material's polarity or roughness, changing the hydrophobicity of the polyamide samples.

- Some of the SEM images seem to be electrically charged. Apparently, no surface metallization was applied on their surface. If so, it is recommended to declare that samples were not metallized in the text, in order to clarify it to the potential readership. In any case, it is suggested to add a nanoscopic very thin layer of Pt or Au at the small pieces observed under SEM next time.

- Figures 6 and 7: dimension is clear because an additional scalebar is inserted at the bottom right corner, but I wonder why a solid white or black background was not included at the scalebars of each picture. They are barely seen due to the contrast, despite included.

- SEM images are quite large (low magnification), was it tried to measure the surface using an optical microscope?

- Is it possible to use an EDS/X to check the composition of the inclusions observed by SEM? This might confirm that they are "non-melted particles arising from polymer degradation" (line 233), or maybe something else.

- Line 131: An explanation of the application of different drying times for PA6 and PA66 (2 and 4 hours, respectively) is recommended.

Best regards.

Author Response

Dear editor and dear reviewers,

Thank you very much for taking the time to review this manuscript. We deeply appreciate your insightful comments and inquiries. Please find the detailed responses below and the corresponding revisions/corrections highlighted in distinct colors to differentiate between the editor's and reviewers' input. Additionally, we have adjusted the title in accordance with the provided recommendations.

Reviewer 4:

  1. UV is known to affect polyamides' properties (for instance, https://doi.org/10.1007/s42452-020-03319-4). Were samples exposed to UV light between each cycle? If not, it is suggested to mention that samples were not exposed to UV light in the manuscript.

Our recycled materials are dedicated to manufacturing assembly components in a finished product. Therefore, their exposure to ultraviolet light was not initially among the considerations. However, to provide clarity to the readers, this observation has been added to the manuscript line 104.

  1. Was hydrophobicity inspected between cycles? Perhaps inclusion and porosity modification on the surface alters material's polarity or roughness, changing the hydrophobicity of the polyamide samples.

Hydrophobicity has been assessed through the measurement of the material's water absorption capabilities. Water absorption tests were conducted on the PA6 and PA66 at various recycling stages and under different conditions. However, the results have been reserved for presentation in another article that will delve more deeply into this subject.

  1. Some of the SEM images seem to be electrically charged. Apparently, no surface metallization was applied on their surface. If so, it is recommended to declare that samples were not metallized in the text, in order to clarify it to the potential readership. In any case, it is suggested to add a nanoscopic very thin layer of Pt or Au at the small pieces observed under SEM next time.

Done line 169 and thank you for your recommendation.

  1. - Figures 6 and 7: dimension is clear because an additional scalebar is inserted at the bottom right corner, but I wonder why a solid white or black background was not included at the scalebars of each picture. They are barely seen due to the contrast, despite included.

Additional scalebar has been meticulously inserted to give clear scale visibility. Thank you for your comments, and we will take them into consideration next time.

  1. - SEM images are quite large (low magnification), was it tried to measure the surface using an optical microscope?

We did not attempt measurements using an optical microscope. Instead, we conducted observations using a scanning electron microscope (SEM) at various magnifications. This choice was made to explore the sample at a microscopic level with higher resolution and to obtain detailed images at different scales. The decision to forego the use of an optical microscope was driven by the need for enhanced magnification capabilities and the ability to investigate the sample's fine structural details. The SEM allowed us to capture images at different levels of magnification, providing a comprehensive view of the sample's features.

  1. Is it possible to use an EDS/X to check the composition of the inclusions observed by SEM? This might confirm that they are "non-melted particles arising from polymer degradation" (line 233), or maybe something else.

We choose not to conduct EDS/X analysis, as our EDS/X is not precise enough. And our research objective was focused on how the recycling process influenced the mechanical and physical characteristics of the polyamide.

Non-melted polymer granules have been analysed in other research papers, for example : https://doi.org/10.3390/polym13234192 .They pointed out that cross-sectional cuts revealed the dispersion of intact polymer granules (un-melted) beneath the film matrices' surface, leading to the formation of non-smooth structures both on the surface and in cross-section.

This reference has been added into the paper line 266.

  1. - Line 131: An explanation of the application of different drying times for PA6 and PA66 (2 and 4 hours, respectively) is recommended.

In the case of condensation polymers such as polyamides, the degradation kinetics are significantly accelerated in the presence of water, which instigates depolymerization reactions. This phenomenon also synergistically interacts with thermomechanical degradation, further accentuating the loss of material properties. Consequently, a careful drying process is imperative prior to the melt operation, a precondition applicable to both PA6 and PA66. Hence, before injection, the PA6 material undergoes a meticulous 2-hour drying regimen at 80°C, while PA66 necessitates 4 hours at the same temperature to eliminate any residual moisture. These specifications are communicated by the supplier and hold UL Recognition (Underwriters Laboratories) that provides a benchmark for safety and performance in relevant applications and they specify a necessary drying if the material have been exposed to air for longer than 3 hours.

Round 2

Reviewer 2 Report (New Reviewer)

Comments and Suggestions for Authors

Dear authors,

Thank you for considering my suggestions.

Reviewer 3 Report (New Reviewer)

Comments and Suggestions for Authors

I accept the authors' explanation.

This manuscript is a resubmission of an earlier submission. The following is a list of the peer review reports and author responses from that submission.

Round 1

Reviewer 1 Report

Comments and Suggestions for Authors

The authors studied the impact of mechanical recycling on the physical and mechanical properties of recycled PA6 and PA66 in relation to their microstructures. They performed a series of experiments and obtained some useful results. The work has a solid engineering background.

The topic is accordance with the target journal. The logic is well planned and the language is roughly smooth. It can be considered for publication after some careful modifications.

1. In the keywords, the format of “microstructure” is not consistent.

2. In L37 of P1, “[5,6,7]” should be “[5-7]”.

3. In the abstract, what is the meaning of “E-modulus”?

4. In Table 3, “Kg” should be “kg”.

5. In many places, “MPA” should be “MPa”.

6. The figures can be further improved by using the stress-strain curve. Their definitions can be given first. Then the Young’s modulus can be got.

7. What is the loading schematic for the bending experiment?

Comments on the Quality of English Language

none

Author Response

Dear reviewers,

Thank you very much for your pertinent comments and questions. We answered point-to point to all your remarks separating the reviewers by colour.

Reviewer 1:

The authors studied the impact of mechanical recycling on the physical and mechanical properties of recycled PA6 and PA66 in relation to their microstructures. They performed a series of experiments and obtained some useful results. The work has a solid engineering background. The topic is accordance with the target journal. The logic is well planned and the language is roughly smooth. It can be considered for publication after some careful modifications.

  1. In the keywords, the format of “microstructure” is not consistent.

The microstructure is changed in Microscopic analysis

  1. In L37 of P1, “[5,6,7]” should be “[5-7]”.

Done

  1. In the abstract, what is the meaning of “E-modulus”?

E-modulus is changed to Young’s modulus

  1. In Table 3, “Kg” should be “kg”.

Done on page 5 line 160

  1. In many places, “MPA” should be “MPa”.

Done: lines 201, 202, 285, 286, Table 2 and Figs. 4a and 4b, 8a et 8c.

  1. The figures can be further improved by using the stress-strain curve. Their definitions can be given first. Then the Young’s modulus can be got.

Regarding the stress-strain curve plotting for all recycling operations. We found that the curves tended to overlap (Please find below the figure), resulting in relatively minor differences between them. Due to this, we decided to focus on plotting the tensile strength, Young's modulus, and elongation at break as a function of the number of recycling processes. This was done to illustrate the evolution of properties observed in PA6 and PA66 as the number of processes increases.

This approach was taken to provide a clearer representation of how these key mechanical properties change over the course of multiple processes, given the limited distinction observed in the stress-strain curves. We believe that this perspective offers a more insightful view of the material behavior under recycling conditions.

  1. What is the loading schematic for the bending experiment?

The 3 points bending test is visible on Figure 2 b.

Reviewer 2 Report

Comments and Suggestions for Authors

Dear Authors,

I was carefully reviewed the submitted article “polymers ID-2674242, title: Repercussions of polyamide recycling on mechanical behavior and shaping by injection”. 

Your research is a very important basic technology for converting to remolded products. This research may provide important information when remolding waste nylon.

The data will be very well summarized and reflect the facts changes of nylon.

But I have many questions;

1. Misspelled title?

2. The temperature during molding is repeated several times (4 to 6), and the nylon naturally undergoes degradation? such as 2 hours at 80°C and 4 hours at the same temperature for PA66. Total treatment time 12 to 24 hours? There is a paper that plastic decomposes at 30.(Appl.Sci.2020,10,5100; doi:103390/app10155100)

3. Where and how the nylon molecule changes! ?(It doesn't match the special issue "latest"! May be use IR, NMR?)

4. Is it possible to evaluate only by the change in color of a sample colored with dye? (Does not match the “latest”) UV or IR?

5. Molecular changes due to GPC are not understood. (Does not match the “latest”)

Analysis by molecular change, GPC is required.

6. Crystal structure changes have not been analyzed by X-ray. (Does not match the “latest”!) Microscopic analysis revealed the following: Imperfections in the crystalline regions of polyamide 6.

I recommend major revision or resubmission as a technical report not article.

Comments on the Quality of English Language

almost OK!

Author Response

Dear reviewers,

Thank you very much for your pertinent comments and questions. We answered point-to point to all your remarks separating the reviewers by colour.

  1. Misspelled title?

The title is changed to:

Repercussions of recycling on the physical and mechanical properties of Polyamides scraps

  1. The temperature during molding is repeated several times (4 to 6), and the nylon naturally undergoes degradation? such as 2 hours at 80°C and 4 hours at the same temperature for PA66. Total treatment time 12 to 24 hours? There is a paper that plastic decomposes at 30℃.(Appl.Sci.2020,10,5100; doi:103390/app10155100)

The mechanical recycling of polymers presents a challenge due to the intricate nature of the process. Repetitive cycles of melt processing can induce subsequent degradation in these materials, leading to subsequent deterioration in their inherent properties. The thermomechanical degradation experienced during melt processing can be mitigated or circumvented by incorporating minute quantities of stabilizers that exert an influence on radical propagation reactions or reactions involving hydroperoxide compounds generated during processing in the presence of oxygen (DOI:10.1002/(SICI)1097-4628(19991017)74:3<510::AID-APP5>3.0.CO;2-6) . Notably, our material already encompasses heat stabilizers, and the thermomechanical degradation is evident as per the outcomes derived from numerous characterization tests delineated in this study.

Also, in the specified study (Appl.Sci.2020,10,5100; doi:103390/app10155100, Section 3.2), They do not mention polyamides and it was elucidated that PS with no additives, such as stabilizers has been shown to degrade under the environmental temperature range.

Furthermore, in the case of condensation polymers such as polyamides, the degradation kinetics are significantly accelerated in the presence of water, which instigates depolymerization reactions. This phenomenon also synergistically interacts with thermomechanical degradation, further accentuating the loss of material properties. Consequently, a careful drying process is imperative prior to the melt operation, a precondition applicable to both PA6 and PA66. Hence, before injection, the PA6 material undergoes a meticulous 2-hour drying regimen at 80°C, while PA66 necessitates 4 hours at the same temperature to eliminate any residual moisture. These specifications are communicated by the supplier and hold UL Recognition (Underwriters Laboratories) that provides a benchmark for safety and performance in relevant applications and they specify a necessary drying if the material have been exposed to air for longer than 3 hours.

PA6 : https://www.ulprospector.com/plastics/fr/datasheet/50910/chemlon-md82

PA66 : https://www.ulprospector.com/plastics/en/datasheet/217089/technyl-a-205f-bk-21n

à An elucidation of this specific material characteristic has been included in the introduction from Line 67 to line 90 and in section 2.2 from line 127 to 130.

  1. Where and how the nylon molecule changes! ?(It doesn't match the special issue "latest"! May be use IR, NMR?)

The viscosity of a polymer melt is influenced by its molecular weight. In Section 3.3, the Melt Flow Rate (MFR) and Melt Volume Rate (MVR) of recycled PA6 and PA66 have been demonstrated. It is well-known that materials with higher molecular weights have more entangled chains, resulting in higher flow resistance. Consequently, a decrease in MFI indicates higher material viscosity, which can be attributed to molecular weight increase and crosslinking-induced degradation. Conversely, an increase in MFI suggests lower polymer viscosity, which arises from chain breakage-induced degradation.

The alterations in nylon molecule characteristics have been elucidated based on the results obtained from MFR and MVR analyses. These analyses provide insights into how changes in molecular weight influence the flow behavior and viscosity of the polymer, shedding light on the underlying mechanisms of degradation and crosslinking processes in recycled PA6 and PA66.

  1. Is it possible to evaluate only by the change in color of a sample colored with dye? (Does not match the “latest”) UV or IR?

Samples colore changes after recycling is already a significant problem have been treated in several research papers:

1-https://doi.org/10.1080/03602559508012182

2-https://doi.org/10.1016/j.polymdegradstab.2021.109748

3-https://doi.org/10.1016/j.matdes.2013.10.037

It was studied in our case but we didn't ditect any changes to present because the vergin material is already colored with ultra-constant color masterbatch uniformly contained at a percentage of 2% which helps to avoid color change after all recycling operations.

à We aim to elucidate this in Section 2.1, spanning from line 111 to line 115.

  1. Molecular changes due to GPC are not understood. (Does not match the “latest”)

Analysis by molecular change, GPC is required.

In this particular paper, our primary focus is on showcasing the alteration of mechanical and physical properties of polyamides 6 and 66 with an increasing number of recycling operations. However, we would like to clarify that our comprehensive study on the blend of virgin and recycled polyamides, aimed at determining the optimal percentage of recycled material for use in the electrical product sector, includes a detailed analysis of molecular changes through GPC analysis. Hence, we have not presented the GPC analysis results in this paper.

  1. Crystal structure changes have not been analyzed by X-ray. (Does not match the “latest”!) Microscopic analysis revealed the following: Imperfections in the crystalline regions of polyamide 6.

The observation of inclusions revealed that their size diminishes as the number of injection cycles for nylon-6 increases. This phenomenon is likely attributed to the presence of unmelted particles resulting from polymer degradation, which serve as nucleation sites, leading to the formation of smaller crystals and subsequently elevating the crystallization temperature.

This analysis and conclusion have been extensively examined in this study (https://doi.org/10.1007/BF02546656) and we used the inclusion size measurements to confirm the analysis done without repeating the whole work.

àWe added this Ref [27] at line 240.

Round 2

Reviewer 2 Report

Comments and Suggestions for Authors

Dear Authors

Authors are not allowed to make amendments to problems in the paper that have been pointed out. Mechanical recycle, which is used as a keyword, is not found anywhere in your paper. It has been pointed out that this paper cannot be published as a professional opinion. However, if the editor of your special issue approves publication,

the content of the article will be the responsibility of the author.

Sincerely yours